# Assessment of Dietary Sodium Intake Using the Scored Salt Questionnaire in Autosomal Dominant Polycystic Kidney Disease

**DOI:** 10.3390/nu12113376

**Published:** 2020-11-02

**Authors:** Annette T. Y. Wong, Alexandra Munt, Margaret Allman-Farinelli, Sunil V. Badve, Neil Boudville, Helen Coolican, Ashley N. Chandra, Susan Coulshed, Mangalee Fernando, Jared Grantham, Imad Haloob, David C. H. Harris, Carmel M. Hawley, Jane Holt, David W. Johnson, Karthik Kumar, Vincent W. Lee, Maureen Lonergan, Jun Mai, Anna Rangan, Simon D. Roger, Sayanthooran Saravanabavan, Kamal Sud, Vicente E. Torres, Eswari Vilayur, Jennifer Q. J. Zhang, Gopala K. Rangan

**Affiliations:** 1Centre for Transplant and Renal Research, Westmead Institute for Medical Research, The University of Sydney, Sydney 2145, Australia; annette.wong@sydney.edu.au (A.T.Y.W.); alexandra.munt@sydney.edu.au (A.M.); ashley.chandra@sydney.edu.au (A.N.C.); david.harris@sydney.edu.au (D.C.H.H.); vincent.lee@sydney.edu.au (V.W.L.); sayan.saravanabavan@sydney.edu.au (S.S.); jennifer.zhang@sydney.edu.au (J.Q.J.Z.); 2Department of Renal Medicine, Westmead Hospital, Western Sydney Local Health District, Sydney 2145, Australia; 3School of Life and Environmental Sciences, The University of Sydney, Sydney 2006, Australia; margaret.allman-farinelli@sydney.edu.au (M.A.-F.); anna.rangan@sydney.edu.au (A.R.); 4St George Hospital, Sydney 2217, Australia; s.badve@unsw.edu.au; 5The George Institute for Global Health, University of New South Wales, Sydney 2042, Australia; 6Sir Charles Gairdner Hospital, Perth 6009, Australia; neil.boudville@uwa.edu.au; 7Medical School, University of Western Australia, Perth 6009, Australia; 8Polycystic Kidney Disease Australia, Roseville 2069, Australia; helen@pkdaustralia.org; 9North Shore Nephrology, Sydney 2065, Australia; s.coulshed@natm.com.au; 10Department of Renal Medicine, Prince of Wales Hospital, Sydney 2031, Australia; Mangalee.Fernando@health.nsw.gov.au; 11Kansas University Medical Center, Kansas City, MO 66103, USA; g.rangan@sydney.edu.au; 12Department of Renal Medicine, Bathurst Hospital, Bathurst 2795, Australia; Imad.Haloob@health.nsw.gov.au; 13Australasian Kidney Trials Network, University of Queensland at Princess Alexandra Hospital, Brisbane 4102, Australia; Carmel.Hawley@health.qld.gov.au (C.M.H.); David.Johnson2@health.qld.gov.au (D.W.J.); 14Translational Research Institute, Brisbane 4102, Australia; 15Department of Nephrology, Princess Alexandra Hospital, Brisbane 4102, Australia; 16Department of Renal Medicine, Wollongong Hospital, Wollongong 2500, Australia; Jane.Holt@health.nsw.gov.au (J.H.); mvero@bigpond.net.au (M.L.); 17Gosford Nephrology, Gosford 2250, Australia; karthik@gosfordnephrology.com; 18Department of Renal Medicine, Liverpool Hospital, Southwestern Sydney Local Health District, Sydney 2170, Australia; Jun.Mai@health.nsw.gov.au; 19Renal Research, Gosford 2250, Australia; sdroger@bigpond.net.au; 20Department of Renal Medicine, Nepean Hospital, Sydney 2751, Australia; Kamal.Sud@health.nsw.gov.au; 21Translational Polycystic Kidney Disease Center, Mayo Clinic, Rochester, NY 55902, USA; torres.vicente@mayo.edu; 22Department of Nephrology, John Hunter Hospital, Newcastle 2305, Australia; Eswari.Vilayur@hnehealth.nsw.gov.au

**Keywords:** autosomal dominant polycystic kidney disease, diet, sodium, salt, progression, 24-h urine, food frequency questionnaire

## Abstract

The excess intake of dietary sodium is a key modifiable factor for reducing disease progression in autosomal dominant polycystic kidney disease (ADPKD). The aim of this study was to test the hypothesis that the scored salt questionnaire (SSQ; a frequency questionnaire of nine sodium-rich food types) is a valid instrument to identify high dietary salt intake in ADPKD. The performance of the SSQ was evaluated in adults with ADPKD with an estimated glomerular filtration rate (eGFR) ≥ 30 mL/min/1.73 m^2^ during the screening visit of the PREVENT-ADPKD trial. High dietary sodium intake (HSI) was defined by a mean 24-h urinary sodium excretion ≥ 100 mmol/day from two collections. The median 24-h urine sodium excretion was 132 mmol/day (IQR: 112–172 mmol/d) (*n* = *75*; mean age: 44.6 ± 11.5 years old; 53% female), and HSI (86.7% of total) was associated with male gender and higher BMI and systolic blood pressure (*p* < 0.05). The SSQ score (73 ± 23; mean ± SD) was weakly correlated with log_10_ 24-h urine sodium excretion (*r* = 0.29, *p* = 0.01). Receiving operating characteristic analysis showed that the optimal cut-off point in predicting HSI was an SSQ score of 74 (area under the curve 0.79; sensitivity 61.5%; specificity 90.0*%; p* < 0.01). The evaluation of the SSQ in participants with a BMI ≥ 25 (*n* = 46) improved the sensitivity (100%) and the specificity (100%). Consumers with an SSQ score ≥ 74 (*n* = 41) had higher relative percentage intake of processed meats/seafood and flavourings added to cooking *(p* < 0.05). In conclusion, the SSQ is a valid tool for identifying high dietary salt intake in ADPKD but its value proposition (over 24-h urinary sodium measurement) is that it may provide consumers and their healthcare providers with insight into the potential origin of sodium-rich food sources.

## 1. Introduction

Autosomal dominant polycystic kidney disease (ADPKD) is the most common (1:1000) adult-onset genetic chronic kidney disease (CKD) caused by variants in either *PKD1*, *PKD2* or, rarely, other genes [1]. The cardinal feature of ADPKD is the formation and growth of multiple kidney cysts, which leads to a fifty percent lifetime risk of developing kidney failure, recurrent episodic kidney pain and hypertension [2,3]. The excess intake of dietary sodium has been identified as a key modifiable factor for promoting the post-natal growth of kidney cysts and hypertension in ADPKD [4]. In one of the largest clinical trials conducted in PKD, the HALT Progression of Polycystic Kidney Disease (HALT-PKD) study comprising 1044 adults with ADPKD, high dietary sodium intake was associated with a greater total kidney volume (TKV) growth rate and a steeper decline in estimated glomerular filtration rate (eGFR) [5,6]. Based on these data, as well as the benefits of the Dietary Approaches to Stop Hypertension (DASH) diet in the general [7] and CKD population [4,8], clinical practice guidelines in ADPKD recommend the restriction of dietary sodium intake to ≤100 mmol/day (<6 g salt or <2.3 g sodium daily) to reduce disease complications [9,10,11,12]. 

However, the majority of adults with ADPKD typically exceed the recommended dietary intake of sodium [4]. The Consortium for Radiologic Imaging Studies of PKD (CRISP) demonstrated that the annual 24-h urine sodium excretion consistently exceeded 190 mmol/day in North American adults with ADPKD over a six-year period [6,13] and informal dietary counselling was insufficient to alter dietary habits [5]. For example, in the HALT-PKD study, instruction by the clinical trial coordinators to reduce salt intake to less than 100 mmol/day led only to a marginal reduction in the urinary sodium excretion of between 6% and 14.5% from baseline mean values (178 mmol sodium/day) [5]. Although persons with ADPKD are probably motivated to implement this intervention, a major barrier may be the absence of simple methods to assess dietary sodium intake [14,15]. Traditionally, the gold-standard biomarker to estimate sodium intake has been twenty-four-hour urine sodium excretion, as approximately 90% of dietary sodium is excreted in urine [16,17,18], but this method is time-consuming and inconvenient for consumers, and there is an increased risk of error in collection [19]. Food frequency questionnaires (FFQ), on the other hand, are a quick and inexpensive method for healthcare practitioners to easily identify high sodium consumers [20]. Although FFQs have potential for systematic error, their relative ease of use might provide a more practical and acceptable method to rapidly identify the intake of high dietary salt [21] in ADPKD [16,19]. 

There has been little research undertaken into the validity and performance of FFQs to estimate dietary salt intake in ADPKD. The Scored Sodium Questionnaire (SSQ) is an FFQ that was developed and validated at the Royal Brisbane and Women’s Hospital in Australia to identify high salt intake in patients with chronic kidney disease stages 3 to 5 [20,22]. The SSQ is a 35-item instrument which categorises into nine key sodium-rich food sources and requests information regarding the use of discretionary salt, dining out and use of salt-reduced products [22]. Therefore, the aim of this study was to evaluate the hypothesis that the SSQ is a valid instrument for identifying high dietary salt intake in ADPKD.

## 2. Materials and Methods 

### 2.1. Study Population

This project is a sub-study of the PREVENT-ADPKD trial registered with the Australian New Zealand Clinical Trials Registry (ANZCTR) (No. ACTRN12614001216606) [23]. All consecutive participants that consented and attended the first study visit (screening visit) of the PREVENT-ADPKD clinical trial between May 2017 and May 2018 were included [23]. Recruitment occurred at thirteen Australian sites including in Sydney, regional New South Wales, Brisbane and Perth [23]. Participants met the inclusion criteria for the PREVENT-ADPKD study, which was a diagnosis of ADPKD, aged between 18 and 67 years old, estimated glomerular filtration rate (eGFR) ≥ 30 ml/min/1.73 m^2^ and providing informed consent. The exclusion criteria were a contraindication to undergo magnetic resonance imaging (MRI) (such as pregnancy, claustrophobia) (as this is the primary outcome measure in the PREVENT-ADPKD study), being considered non-compliant, a confounding illness or treatment and/or participation in another clinical trial [23]. The study was conducted according to the Declaration of Helsinki and approved by the Western Sydney Local Health District (WSLHD) Human Research Ethics Committee (HREC), HREC ref: AU RED HREC/14/WMEAD/414.

### 2.2. Study Design

Data were collected at the screening visit and included height, body weight, body mass index (BMI) and office blood pressure, as described in the PREVENT-ADPKD trial protocol [23]. During the visit, participants self-completed the SSQ and were instructed to perform two 24-h urine collections for urine sodium excretion and a blood test for serum creatinine to calculate the eGFR over the following 12 weeks. 

### 2.3. Measurement of Urinary Sodium Excretion

The analysis of serum creatinine and 24-h urine sodium was performed by a clinical pathology laboratory (Douglass Hanly Moir, Ryde, NSW, Australia) accredited by the National Association of Testing Authorities, Australia. The eGFR was determined by the Chronic Kidney Disease Epidemiology Collaboration (CKD-EPI) formula. The dietary sodium intake was determined from the mean of the two 24-h urine sodium excretions [19] and high sodium intake was defined as the mean 24-h sodium excretion that exceeded 100 mmol/day [9]. Incomplete collection of a 24-h urine collection was defined by a creatinine index (urinary creatinine (mmol/day) × 113)/(21 × body weight (kg)) of less than 0.7 [24,25,26] whereas overcollection if the 24-h urine creatinine excretion was >25 mg/kg/day in males or >20 mg/kg/day in females [27,28]. Participants with at least one sample meeting these criteria were excluded from further analysis.

### 2.4. Scored Salt Questionnaire

The SSQ was a self-reported questionnaire that determines the frequency of the usual consumption of sodium-rich food types over the previous six months [22]. The food types were categorised into nine groups and scored based on frequency of consumption (either never/rare; once per week; 2–3 times per week or at least once daily) [22]. The groups and the maximum score (MS) for each were as follows: (*i*) Group 1: Breads (MS = 30); (*ii*) Group 2: Spreads (MS = 12); (*iii*) Group 3: Cereals, biscuits and baking (MS = 17); (*iv*) Group 4: Cheeses and savory snacks (MS = 20); (*v*) Group 5: Tinned and packed foods and other meal components (MS = 19); (*vi*) Group 6: Processed meats and seafood (MS = 35); (*vii*) Group 7: Flavorings added in cooking (MS = 43); (*viii*) Group 8: Flavorings added at the table (MS = 24); (*ix*) Group 9: Pre-prepared meals and drinks and eating out (MS = 15). The total SSQ score was determined by the sum of individual values for each food type category, with a maximum possible score of 215, with larger values indicative of higher salt consumption [22]. 

### 2.5. Statistical Analysis

Statistical analysis was conducted using JMP® Pro software (v14.2.0, SAS Institute, Cary, NC, USA) and a *p* value of less than 0.05 was defined as statistical significance. The normality of the distribution was determined using the Shapiro–Wilk W Test. All normally distributed continuous variables were reported as means ± SD, non-normally distributed variables were expressed as median and interquartile range (IQR) and categorical data were reported as *n* (%). Non-normally distributed data were natural log (In)-transformed, as appropriate. Differences between groups were determined using one-way ANOVA, Kruskal–Wallis and Chi-squared test for normally distributed continuous, non-normally distributed and categorical variables, respectively. The 24-h urine sodium excretion was determined from the mean values derived from two collections. The SSQ score was analysed as a continuous variable and the relative contributions of the nine categories were expressed as a percentage of the total score. The correlation between SSQ score and log 24-h urine sodium excretion was determined using the Spearman correlation coefficient. The receiver operating characteristic (ROC) analysis and measures of diagnostic accuracy were used to determine the performance of the SSQ to predict a 24-h urine sodium excretion of greater than 100 mmol/day.

## 3. Results

### 3.1. Characteristics of High and Low Sodium Intake Groups 

As shown in Figure 1, one hundred and sixty-four participants were invited to complete the SSQ and two 24-h urine collections. Fifty-five participants were excluded due to not completing either the SSQ and/or two urine collections, leaving 109 participants. Of the 109 participants, a further 34 were excluded due to a 24-h urine sample being incomplete (under-collection, *n* = 21) or being over-collected (*n* = 13) (Figure 1). Participants with an under-collected urine sample were more likely to be female and have a higher BMI, whereas those who over-collected had a higher 24-h urine sodium excretion, as expected (*p* < 0.01, Appendix A). 

In participants with complete urine collections, the median 24-h urinary sodium excretion was 132 mmol/day (IQR, 112–172 mmol/day) (Table 1). The majority (86.7%) of the cohort had a 24-h urinary sodium level ≥ 100 mmol/day, and these participants were more likely to be male and have a higher body weight, height, BMI and systolic blood pressure than those with 24-h urinary sodium < 100 mmol/day (Table 1). The median number of days between the two 24-h urine collections was 14 (IQR, 6.75–14 days), and this was not correlated with the urinary sodium excretion (*r* = 0.035, *p* = 0.77). Participants performed the majority (88%) of both urine collections on weekdays. 

### 3.2. Performance of the SSQ with 24-H Urine Sodium Excretion

The mean SSQ score of the study population was 73 ± 23 and was lower in participants with 24-h urine Na < 100 mmol/day (24-h urine Na < 100 mmol/day: 53 ± 21; 24-h urine Na ≥ 100 mmol/day: 76 ± 22; *p* < 0.01). The relationship between the SSQ score and the log mean 24-h urine sodium excretion was weak (r = 0.29; *p* = 0.01) (Figure 2). By multivariable analysis (gender, height, BMI, weight, systolic blood pressure and the SSQ score), only the BMI and SSQ score were significantly associated with the log mean 24-h urine sodium excretion (*p* < 0.01). By the ROC analysis, an SSQ score of 74 was the optimal cut-off point to predict 24-h urine Na ≥ 100 mmol/day, with an area under the curve (AUC) of 0.79 (*p* < 0.01) (Figure 3A). The ROC analysis for BMI alone demonstrated that a value of 24.7 kg/m^2^ was the optimal cut-off point to predict 24-h urine Na ≥ 100 mmol/day, with an AUC of 0.75 (*p* = 0.02) (Figure 3B). The performance characteristics of the SSQ and BMI are shown in Table 2. Furthermore we evaluated the performance of the SSQ in participants with BMI ≥ 25 kg/m^2^ (*n* = 46) and <25kg/m^2^ (*n* = 29) and the results showed that the accuracy improved to 100% in the former (*p* < 0.01) but was not significant in the latter group (AUC = 0.72, *p* = 0.13) (Table 2). 

### 3.3. Characteristics of the Sodium-Rich Foods Types Identified by the SSQ

The breakdown of the SSQ score into each of the nine sodium-rich food types for the study population, as determined by the absolute values, is shown in Figure 4A and Appendix A. The top five sodium-rich food groups as a median percentage of the total SSQ score were: Group 7 (Flavorings added in cooking; 19.5%), Group 1 (Breads; 19.0%), Group 6 (Processed meats and seafood; 17.0%); Group 8 (Flavorings added at the table; 9.4%) and Group 4 (Cheeses and Savory Snacks; 9.1%) (Appendix A). With the exception of Group 3 (Cereals, biscuits and baking), participants with an SSQ ≥ 74 had higher absoluate values for all sodium-rich food categories compared to those with an SSQ < 74 (Figure 4B and Appendix A). Compared with the individuals scoring SSQ < 74, the median SSQ scores in Group 6 (Processed Meat and Seafood), Group 7 (Flavoring added in cooking) and Group 8 (Flavorings added at the table) were approximately 3.1-, 2.8- and 2.0-fold higher in those with an SSQ ≥ 74 (Figure 4B and Appendix A). The relative percentage contribution to the total SSQ score from Group 1 (Breads) was the only category lower in the SSQ ≥ 74 group compared to the SSQ score < 74 group whereas Group 3 (cereal), Group 4 (Cheeses) and Group 6 (Processed Meat) and 7 (Flavourings) were higher (Appendix A), suggesting preferential differences in dietary sources of salt between the two groups.

## 4. Discussion

This study has evaluated the performance of a food frequency questionnaire to screen for high dietary salt intake in people with ADPKD. The key findings were: (*i*) a score of greater than 74 from the SSQ predicted high dietary salt intake (as defined by 24-h urine sodium excretion of ≥ 100 mmol/day) with a PPV of 97.6% and a NPV of 26.5%, and both of these parameters improved to 100% in participants who had a BMI ≥ 25; (*ii*) the intake of specific sodium-rich food groups (Processed meats/seafood and flavourings added in cooking or at the table) made a higher contribution to the SSQ score in participants with an SSQ ≥ 74; and (*iii*) the 24-h urinary sodium excretion was a burdensome method to assess dietary salt, given that 31.2% of our population made errors during the collection period and had to be excluded from the analysis. Taken together, these data suggest that the SSQ is a valid instrument for identifying high dietary sodium intake in ADPKD but its unique value proposition (over 24-h urine collection) is that it is likely to provide insight to both consumers with ADPKD as well as their healthcare practitioners into the potential origin of sodium-rich food sources.

The results of the current study demonstrate that the SSQ is a simple instrument for identifying high dietary salt intake in the ADPKD population [22]. In previous studies, Ross et al. reported that the SSQ could be self-completed in an outpatient setting with a median completion time of 10 min [20]. In the current study, we also found that participants completed the SSQ in a similar timeframe during outpatient visits with minimal assistance from staff. Because the SSQ was first developed in people with CKD, the main goal of this study was to determine its performance in ADPKD, and we found that there were some slight differences. First, the correlation between the SSQ score and 24-h urine sodium excretion in our study (*r* = 0.29; *p* = 0.01) was lower than that reported by Mason et al. (*r* = 0.371, *p* = 0.031) [22], but not dissimilar to that achieved by other salt food frequency questionnaires [29]. Second, we found that the optimal cut-off point for the SSQ score was 74, with a PPV of 97.6% and NPV of 26.5%. In contrast, Mason et al. reported that a score of 65 was best for their study population, with a PPV of 88% and a NPV of 44% [22]. These slight differences could be due to variations between the two study populations, as the cohort in the study by Mason et al. was older (69.7 ± 13.5 years old), with CKD stages 3–5 [22], compared to the population in the current report (age 44.6 ± 11.5 years old, with CKD stages 1–3). In our study, we also found that higher BMI was associated with high dietary salt intake, and for BMI ≥ 25 kg/m^2^, the ROC analysis improved the global performance of the SSQ, with the AUC in the ROC curve increasing from 0.79 to 1.00. In contrast, we found that the SSQ performed poorly in those with BMI < 25kg/m^2^.

The excess intake of dietary salt is a major problem in the ADPKD population because of its impact on long-term disease outcomes [5]. The implementation of dietary salt restriction in clinical practice has been challenging, and this is reflective of the reliance and pervasive availability of processed and sodium-rich foods in the broader community [30,31]. As noted in the present study, 86.7% of people with ADPKD were categorised as having a high dietary salt intake. Although high quality evidence supports the efficacy of dietary sodium restriction for improving blood pressure and proteinuria in CKD, and on reducing renal disease progression (cyst growth and eGFR decline) in ADPKD [5], there is poor consumer adherence to clinical practice guidelines [32,33,34,35]. The reasons for this are multifactorial and include not only gaps in dietary sodium knowledge but also personal attitudes and behaviours, such as taste preferences, willpower, socioeconomic factors, social expectation and disease concern [30,36]. In this regard, we hypothesise that young people with ADPKD (<35 years old) have less intrinsic motivation to change behaviour, due to a lack of symptoms, than older people with ADPKD [37]. While the SSQ may not solve all the issues related to poor adherence, we postulate that its use in clinical practice might have two beneficial consequences. First, it may provide individuals with a tool to become self-aware of the importance of their salt intake and foods rich in sodium. Second, it may assist healthcare practitioners in identifying and discussing high salt consumption, which could facilitate referral to dietitians to provide targeted education and counselling. Further clinical audit studies are needed to investigate this hypothesis. 

Not surprisingly, the current study provided further evidence that 24-h urine collection to estimate dietary salt intake is inconvenient for participants in either a research setting and/or in clinical practice, as shown previously [21,25]. In the current study, 31.2% of patients either under-collected their 24-h urine sample (19.3% of total population, as defined by a creatinine index of less than 0.7) [24] or defined as overcollection (11.9% of the total population), and this high prevalence rate is comparable to previous studies [27,38]. Para-aminobenzoic acid (PABA) [39] can be used to assess under-collection [19] but was not performed in the current study due to the complexity and burden to participants [25], and this method also does not identify over-collection [27]. The inconvenience of 24-h collection contrasts with the SSQ, which, despite its limitations, is far easier to perform in the clinical setting. However, it will be important in future studies to evaluate whether additional biomarkers such as copeptin [40] and spot urine sodium [19] are suitable surrogate indicators with good diagnostic performance to screen for excess dietary salt intake in ADPKD, in combination with the SSQ.

The strengths of the present study are the sample size, use of multiple statistical methods, assessment of urine completeness (both over- and under-collected samples excluded to reduce bias) and that two 24-h urine collections per participant was used as the reference method to estimate dietary sodium intake [29]. There are several limitations. First, the version of the SSQ used in this study has been designed for a population consuming a western diet and might not apply to other cultures with different eating patterns [41]. Thus, modification of the wording of the SSQ and the type of sodium-rich food will be needed to determine its utility in ADPKD populations living in other countries. Second, the SSQ (like all food frequency questionnaires) is subject to recall bias and under-reporting, which could lead to underestimation of sodium intake. Third, the results of this study are based on participants recruited to a clinical trial, which might influence their dietary behaviours and questionnaire responses and may not be representative of the entire ADPKD population. It would also be important to address the hypothesis of this study in children, such as at-risk adolescents [2]. Fourth, the sample sizes of the two groups were unequal; therefore, ideally, larger cohort studies and/or clinical trials should be undertaken using the SSQ in the future to provide additional evidence for its use in ADPKD. Lastly, participants performed the majority of the 24-h urine collections on weekdays (88%) but data on whether these were working days were not determined, as this could influence dietary sodium intake behaviour.

In conclusion, the results of the study show that the SSQ is a valid instrument for screening for high dietary salt intake in the ADPKD population. Further studies should be undertaken to evaluate whether the combination with other biomarkers, such as the estimation of spot urine sodium, further enhances the diagnostic performance of the SSQ to assess dietary salt intake in ADPKD [21]. At a fundamental level, the use of the SSQ in clinical practice will provide a focused approach for initiating a dialogue with people with ADPKD about dietary salt intake and encouraging changes in behaviour. In this regard, another key question regarding the SSQ is whether its longitudinal use in people with ADPKD can be effective in raising “salt consciousness” [21] and lead to a sustained reduction in dietary sodium intake in individuals [36]. Future cohort studies are needed to address the impact of these simple interventions on disease outcomes in ADPKD.

## Figures and Tables

**Figure 1 nutrients-12-03376-f001:**
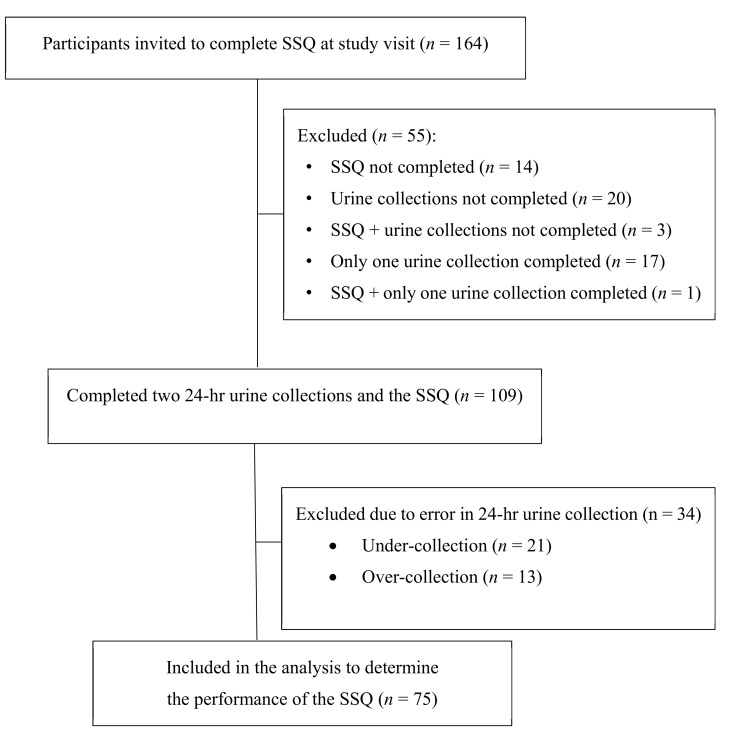
Flow diagram showing the number of participants recruited, excluded and included in the study. Abbreviation: SSQ, scored salt questionnaire.

**Figure 2 nutrients-12-03376-f002:**
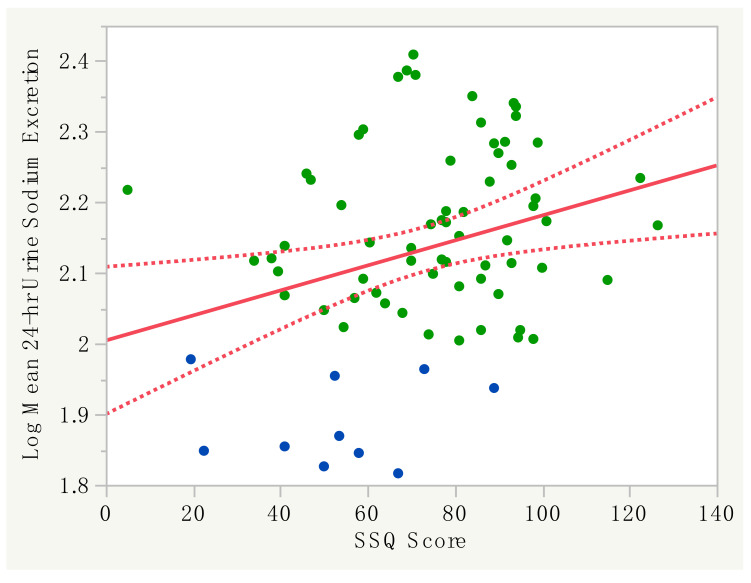
Graph showing the correlation between the SSQ score and log mean 24-h urine sodium showing the line of fit (solid line) (*r* = 0.29; *p* = 0.01) and confidence curves for the fitted line (dotted line and shaded area). Green dots represent participants with 24-h urine Na ≥ 100 mmol/day and blue dots with Na < 100 mmol/day.

**Figure 3 nutrients-12-03376-f003:**
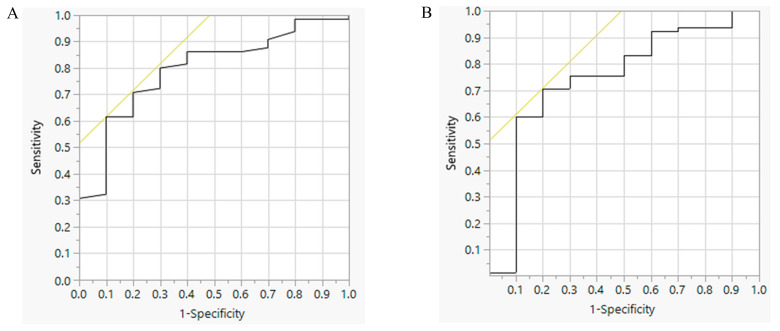
Performance of the SSQ score to predict 24-h urine sodium excretion ≥ 100 mmol/day using receiving operating characteristic (ROC) analysis. Panel (**A**) shows ROC curve with the SSQ score alone (AUC 0.79, *p* < 0.01) and Panel (**B**) shows the ROC curve for BMI alone (AUC 0.75, *p* = 0.02).

**Figure 4 nutrients-12-03376-f004:**
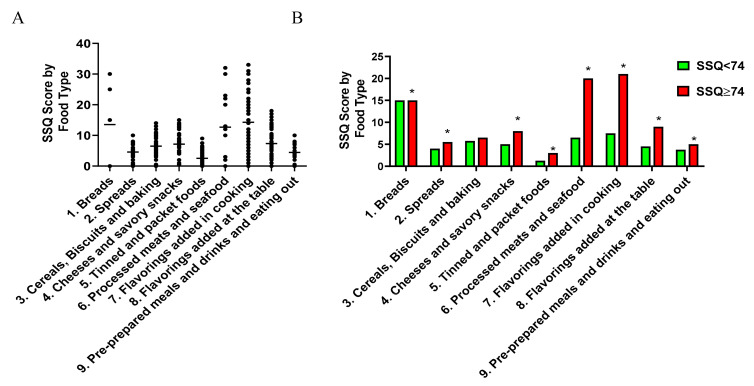
Characteristics of sodium-rich food types in the study population according to absolute SSQ score values. Panel (**A**) shows the absolute scores for each sodium-rich food group for the total study population (*n* = 75) and Panel (**B**) shows the median percentage scores divided into groups that had SSQ score < 74 (green) (*n* = 34) and SSQ score ≥ 74 (red) (n = 41). * *p* < 0.05 compared to SSQ < 74.

**Table 1 nutrients-12-03376-t001:** Clinical characteristics of the high and low sodium intake groups in the study population.

Parameter	Total *n* = 75	24-h Urine Na < 100 mmol/day *n* = 10	24-h Urine Na ≥ 100 mmol/day *n* = 65	*p*
Age (years)	44.6 ± 11.5	40.2 ± 12.5	45.3 ± 11.3	0.19
Gender (F:M) (%)	53:47	90:10	48:52 *	<0.01
Height (m)	1.71 ± 0.09	1.65 ± 0.08	1.72 ± 0.09 *	0.04
Weight (kg)	76.1 ± 15.1	64.1 ± 11.9	78.0 ± 14.8 *	<0.01
BMI (kg/m^2^)	25.9 ± 3.8	23.5 ± 4.4	26.3 ± 3.6 *	0.03
Systolic BP (mm Hg)	133 ± 14	125 ± 10	134 ± 14 *	0.04
Diastolic BP (mm Hg)	85 ± 11	79 ± 8	86 ± 11	0.08
Serum Cr (mmol/L)	91 (70–123)	77 (62–104)	94 (70–126)	0.12
eGFR (mL/min/1.73m^2^)	75 (53–105)	85 (56–112)	73 (52–104)	0.39
24-h urine Na (mmol/day)	132 (112–172)	73 (69–91)	142 (123–180) *	<0.01

Abbreviations: Na, sodium; BMI, body mass index; BP, blood pressure; Cr, creatinine; eGFR, estimated glomerular filtration calculated using the Chronic Kidney Disease Epidemiology Collaboration equation. Data expressed as mean ± SD, with exception of serum Cr and eGFR, which are presented as median with interquartile range; * *p* < 0.05 compared to 24-h urine Na < 100 mmol/day.

**Table 2 nutrients-12-03376-t002:** Performance characteristics of the SSQ and the BMI in predicting high dietary salt intake, as determined by 24-h urine Na ≥ 100 mmol/day.

Parameter	TP	FP	TN	FN	Sensitivity	Specificity	PPV	NPV	LR+	LR−	DOR	A	AUC	*p*
SSQ	40	1	9	25	61.5%	90.0%	97.6%	26.5%	6.15	0.43	14.40	65.3	0.79	<0.01
BMI	46	2	8	19	70.8%	80.0%	95.8%	29.6%	3.54	0.37	9.68	72.0	0.75	0.02
SSQ in BMI < 25 Group (*n* = 29)	12	1	7	9	57.0%	87.5%	92.3%	43.8%	2.87	0.20	14.52	65.5	0.72	0.13
SSQ in BMI ≥ 25 Group (*n* = 46)	44	0	2	0	100%	100%	100%	100%	-	0.00	-	100%	1.00	< 0.01

Abbreviations: TP, true positive; FP, false positive; TN, true negative; FN, false negative; PPV, positive predictive value; NPV, negative predictive value; LR+, positive likelihood ratio; LR-, negative likelihood ratio; DOR, diagnostic odds ratio; A, accuracy; AUC, area under curve.

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
