# Peer review of "Assessment of Dietary Sodium Intake Using the Scored Salt Questionnaire in Autosomal Dominant Polycystic Kidney Disease"

_nutrients, 2020, doi:10.3390/nu12113376_

Round 1
Reviewer 1 Report
The authors used the SSQ, which was validated in a different clinical population, in patients with ADPKD. They correlated results with two 24h urine collections which were collected within 12 weeks of patients' completion of the SSQ. The paper is well-written. Statistical methods appear sound. I have 2 questions of minor import:
1) The SSQ was administered in clinic, and then the subject completed two 24h urine collections over the following 12 weeks. What was the time between collections? Were they days apart or weeks apart? Was it different between subjects? And was there any evidence that a longer period of time between collections was more (or less) reflective of habitual salt intake, as captured by the SSQ?
2) Were working and non-working days included? It is well known that people eat differently on days off of work (typically weekend days) vs. working days. If people completed their two 24h urine collections during week days, when salt intake may have been lower (e.g., due to eating out less often, for example), how does this affect your interpretation of the results?
If the authors are unable to address these questions in a review of their statistical analysis, then perhaps they could comment on these in their discussion and/or limitations.
Author Response
1) The SSQ was administered in clinic, and then the subject completed two 24h urine collections over the following 12 weeks. What was the time between collections? Were they days apart or weeks apart? Was it different between subjects? And was there any evidence that a longer period of time between collections was more (or less) reflective of habitual salt intake, as captured by the SSQ?
Response: The interval between the two 24hr urine collections varied for each individual. The median was 14 days (IQR: 6.75-21). A correlation analysis showed that the number of days between collection and the mean 24-hr urine Na has no significant relationship (r=0.035, p=0.77). We have updated this on Page 4, lines 173-175.
2) Were working and non-working days included? It is well known that people eat differently on days off of work (typically weekend days) vs. working days. If people completed their two 24h urine collections during week days, when salt intake may have been lower (e.g., due to eating out less often, for example), how does this affect your interpretation of the results?
If the authors are unable to address these questions in a review of their statistical analysis, then perhaps they could comment on these in their discussion and/or limitations.
Response: Most of the 24-hr urine collections were collected on weekdays (88% of both collections), however we have not collected details of whether these were working days or not. We have included these data on Page 4, line 175, and mentioned this as a limitation in the Discussion (Page 10, lines 306-308).
Reviewer 2 Report
Overview: The majority of adults with Autosomal Dominant Polycystic Kidney Disease (ADPKD) typically exceed the recommended dietary intake of sodium, which identifies as a key modifiable factor for promoting the post-natal growth of kidney cysts and hypertension. Little research has been undertaken into the validity and performance of Food Frequency Questionnaires (FFQs) to estimate dietary salt intake in ADPKD. In this article Wong A.T.Y. et al., evaluated the hypothesis that the Scored Sodium Questionnaire (SSQ), a 35-item instrument which categorizes key sodium-rich food sources and requests information regarding the use of salt, is a valid instrument for identifying high dietary salt intake in ADPKD. They were able to show that after the 24-hr urine collections, the median 24-hr urinary sodium excretion was 132 mmol/day, which is higher than the recommended dose of ≤100 mmol/day to reduce disease complications. In addition, the majority of the cohort had a 24-hr urinary sodium ≥100 mmol/day, with participants more likely to be male and have a higher body weight, height, BMI and systolic blood pressure than those with 24-hr urinary sodium <100 mmol/day. A multivariate analysis showed that only the BMI and SSQ score were significantly associated with the log mean 24-hr urine sodium excretion. A ROC analysis demonstrated that an SSQ score of 74 was the optimal cut-point to predict 24-hr urine Na≥100 mmol/day. The top five sodium-rich food groups as a median percentage of the total SSQ score were: Group 7 (Flavorings added in cooking; 19.5%), Group 1 (Breads; 19.0%), Group 6 (Processed Meat and Seafood; 17.0%); Group 8 (Flavorings added at the Table; 9.4%) and Group 4 (Cheeses and Savory Snacks; 9.1%), with the exception of Group 3 (Cereals, biscuits and baking). The median scores in Group 6 (Processed Meat and Seafood), Group 7 (Flavoring added in cooking) and Group 8 (Flavorings added at the Table) were approximately 3.1-, 2.8- and 2.0-fold higher in individuals with an SSQ≥74 compared to those with an SSQ<74. In conclusion, Wong A.T.Y. et al. demonstrated that the SSQ is a simple and valid instrument for identifying high dietary sodium intake in ADPKD population. Overall, the scientific article is well written and described, the results are interesting and explicative. All of the tables and figures used are appropriate in presenting the results. However, major and minor changes are suggested:
Major comments:
- In Supplemental Table 1, the number of participants in the ‘Over-collection of 24-hr Urine’ does not add up to 100% of Gender (F:M), the amount given in Table S1 is 62:47, which adds to 109%. This should be corrected, and will this correction change the values for all other measurements?
- In Table 1, the number of participants in the ‘24-hr urine Na ≥100 mmol/day’ does not add up to 100% of Gender (F:M), the amount given in Table 1 is 48:55, which adds to 103%. This should be corrected, and will this correction change the values for all other measurements?
- When comparing the 24-hr urine Na ≥100 mmol/day values to the 24-hr urine Na <100 mmol/day in Table 1, the majority of 24-hr urine Na <100 mmol/day participants (90%) are female, will the results change if more males were added to the 24-hr urine Na <100 mmol/day cohort?
- Why was the creatine excretion values from the 24hr-urine collection, to exclude participants as mentioned in Page 3 lines 127-130, were not added in Table S1? The Table S1 is showing Serum Creatine, which would be acceptable values to include in the study.
- In Figure 4B, why is Category 1.Breads showing a statistical significance, when there is no visible difference between SSQ<74 and SSQ≥74 groups? The Table S2 is also showing both groups with absolute value of 15.
- In Table S2, verify all Median values for the SSQ Score by Food Type (SSQ<74) group, as well as some of the Median values for the other two groups. Is other information considered? The same observation for Table S3. After revising all Median values does the significance difference changes or stays the same?
- What is the following statement in Page 7 lines 217-220 describing? Other group values in Table S3 are also lower in the SSQ≥74 compared to SSQ<74 group. This statement is not clear.
Minor comments:
- In the authors list, is Helen Coolican affiliated with 6North Shore Nephrology, Sydney, Australia? This information is missing.
- The page numbers are re-starting after Page 4.
- In the introduction section, does HALT stand for something specific?
- A sentence in the introduction section can be added to describe the importance of measuring the estimated glomerular filtration rate (eGFR), which is not always know.
- In the Materials and Methods, the Scored Salt Questionnaire section (Page 4, lines 142-143), why was 215 the maximum possible score for the sum of individual values for each food type category? When the total SSQ score for a food type category doesn’t go higher than MS=43, and the total of all categories is MS=199.
- In the discussion section, Page 9 line 277,the percentage of under-collected patients (n=21) compared to total population (n=109) should be corrected.
Author Response
Major comments:
- In Supplemental Table 1, the number of participants in the ‘Over-collection of 24-hr Urine’ does not add up to 100% of Gender (F:M), the amount given in Table S1 is 62:47, which adds to 109%. This should be corrected, and will this correction change the values for all other measurements?
Response: Thank you for spotting this error. We have revised and corrected the data to 38:62.
- In Table 1, the number of participants in the ‘24-hr urine Na ≥100 mmol/day’ does not add up to 100% of Gender (F:M), the amount given in Table 1 is 48:55, which adds to 103%. This should be corrected, and will this correction change the values for all other measurements?
Response: Thank you for spotting this error. We have revised and corrected the data to 48:52.
- When comparing the 24-hr urine Na ≥100 mmol/day values to the 24-hr urine Na <100 mmol/day in Table 1, the majority of 24-hr urine Na <100 mmol/day participants (90%) are female, will the results change if more males were added to the 24-hr urine Na <100 mmol/day cohort?
Response: This is an interesting question. By multivariable analysis (page 6, line 186), gender was not associated with 24-hr urine sodium excretion, therefore more males in the 24-hr urine Na <100 mmol/day group is not likely to change the outcomes of this study.
- Why was the creatine excretion values from the 24hr-urine collection, to exclude participants as mentioned in Page 3 lines 127-130, were not added in Table S1? The Table S1 is showing Serum Creatine, which would be acceptable values to include in the study.
Response: The 24hr urine creatinine excretion values have been added to Table S1.
- In Figure 4B, why is Category 1.Breads showing a statistical significance, when there is no visible difference between SSQ<74 and SSQ≥74 groups? The Table S2 is also showing both groups with absolute value of 15.
Response: These data have been verified and are all correct. Although the absolute median values are the same for both SSQ<74 and SSQ≥74 groups, statistical significance is confirmed (Table S2). This can be explained by the difference in the distribution of the scores as shown in Table S2, where the IQR are different SSQ<74 group (15-15) vs. SSQ≥74 (0-15).
- In Table S2, verify all Median values for the SSQ Score by Food Type (SSQ<74) group, as well as some of the Median values for the other two groups. Is other information considered? The same observation for Table S3. After revising all Median values does the significance difference changes or stays the same?
Response: These data have been verified and are all correct. All significant differences have remained the same.
- What is the following statement in Page 7 lines 217-220 describing? Other group values in Table S3 are also lower in the SSQ≥74 compared to SSQ<74 group. This statement is not clear.
Response: The sentence has been revised to “Compared with the individuals scoring SSQ<74, the median scores in Group 6 (Processed Meat and Seafood), Group 7 (Flavoring added in cooking) and Group 8 (Flavorings added at the Table) were approximately 3.1-, 2.8- and 2.0-fold higher than those with an SSQ≥74 (Figure 4B and Supplementary Table 2).”
Minor comments:
- In the authors list, is Helen Coolican affiliated with 6North Shore Nephrology, Sydney, Australia? This information is missing.
Response: Thank you for spotting this error. Helen Coolican is affiliated with PKD Australia
- The page numbers are re-starting after Page 4.
Response: Thank you for spotting this error. Multiple authors have attempted to correct this on times however oddly the correction does not appear to save when the document is closed. We kindly refer to the Editorial Team to provide us with assistance on this.
- In the introduction section, does HALT stand for something specific?
Response: HALT is a large Polycystic Kidney Disease study comprising 1,044 adults with ADPKD, high dietary sodium intake was associated with a greater total kidney volume (TKV) growth rate and a steeper decline in estimated glomerular filtration rate (eGFR). HALT stands for (HALT progression of Polycystic Kidney Disease) Study. We have updated this on Page 2, line 68.
- A sentence in the introduction section can be added to describe the importance of measuring the estimated glomerular filtration rate (eGFR), which is not always know.
Response: The importance of eGFR in ADPKD has been mentioned on Page 2, line 69-70.
- In the Materials and Methods, the Scored Salt Questionnaire section (Page 4, lines 142-143), why was 215 the maximum possible score for the sum of individual values for each food type category? When the total SSQ score for a food type category doesn’t go higher than MS=43, and the total of all categories is MS=199.
Response: Thank you for spotting this error. We have corrected the MS for Group 6 from 19 to 35 which confirms the total SSQ score is 215.
- In the discussion section, Page 9 line 277, the percentage of under-collected patients (n=21) compared to total population (n=109) should be corrected.
Response: Thank you for spotting this error. We have revised and corrected the data to 19.3%.
Reviewer 3 Report
This study show that the SSQ is valid instrument for screening for
300 high dietary salt intake in the ADPKD population.
- The partecipants with 24-hr urine Na ≥100 mmol/day had a blood pressure higher than the group with Na< 100 mol/day. Can you comment this result?
- Have they any biochemical data such as renin and aldosterone?
- The number of partecipants with Na< 100 mol/day is too small in comparison to the other group. Therefore, the authors should include more subjects to support the statistical analysis.
-
Please indicate with a different symbol the partecipants with Na< 100 mol/dayhe in the graph showing the correlation between the SSQ score and log mean 24-hour urine sodium, in this way, we can understand better this type of association.
-
It is no clear the reason to add T BMI to the SSQ score to predict 24-hr urine Na≥100 to improve the performance of ROCcurve. This result has to be discussed.
Author Response
- The participants with 24-hr urine Na ≥100 mmol/day had a blood pressure higher than the group with Na< 100 mol/day. Can you comment this result?
Response: Although systolic blood pressure was elevated in the high sodium intake group, multivariable analysis showed that only BMI and the SSQ score were associated with high dietary sodium intake (page 6, lines 188-190). Therefore, in this observational cohort study we did not place further weight on the importance of these data.
- Have they any biochemical data such as renin and aldosterone?
Response: Thank you for raising this point. We agree that measurement of renin and aldosterone would provide further pathophysiological insight into the role of the renin angiotensin system. However, we feel that it is beyond the scope of the present study which was designed primarily to evaluate the validity of the SSQ instrument.
- The number of participants with Na< 100 mol/day is too small in comparison to the other group. Therefore, the authors should include more subjects to support the statistical analysis.
Response: Thank you for raising this question. As this was an observational cohort study design, sample sizes between the groups were unequal and it is not possible to include more subjects into the statistical analyses. therefore the SSQ should be further evaluated in prospective cohort studies and clinical trials. This comment has been added to the limitations section in the discussion (page 10, lines 305-306) in the revised manuscript.
- Please indicate with a different symbol the participants with Na< 100 mol/day in the graph showing the correlation between the SSQ score and log mean 24-hour urine sodium, in this way, we can understand better this type of association.
Response: On Figure 2, we have now identified those with Na<100mmol/day as blue dots and green dots for those with 24-hr urine Na≥100mmol/day.
- It is no clear the reason to add T BMI to the SSQ score to predict 24-hr urine Na≥100 to improve the performance of ROCcurve. This result has to be discussed.
Response: Thank you for raising this point. By multivariable analysis, only BMI and the SSQ score were associated with high dietary sodium intake (page 6, lines 188-190). Therefore, we further analysed the ROC curve with BMI taken into consideration.
Reviewer 4 Report
Result
In patients with BMI>25, the SSQ has been shown to have a higher diagnostic accuracy about high sodium intake. How about BMI<25 ?
Is SSQ a predictive tool limited to ADPKD patients with higher BMI ?
The diagnostic accuracy should be demonstrated in patients with BMI lower than 25.
Discussion
The authors compare their results with the report about chronic kidney disease(Mason et al.). However, the study by Mason et al. covered patients with chronic kidney disease stages 3 to 5, whereas the current report covered stages 1 to 3.
The impact of the difference should be mentioned.
Author Response
In patients with

Round 2
Reviewer 2 Report
Many of the first revision comments where address, whoever there are still minor comments:
- Figure 2 is duplicated in Page 6 and Page 7. The article should keep Figure 2 from Page 7, since the blue dots are important to distinguish between both groups. Are the units described in the legend for Figure 2 in Page 7 lines 209-210 correct? Green dots 24-hr urine Na≥100 mmol/d and blue dots with Na <100 mmol/L.
- Figure 3 is of poor quality and cannot distinguishing the numbers. Which figures will be used in the final manuscript, top or bottom panel? Why are Fig 3B top and Fig 3B bottom different?
- Figure 4 is of poor quality and cannot distinguishing the words.
- The statement of Page 9 lines 235-238 is still not clear. Other groups, besides Group 1, are also lower in persons with a SSQ≥74 compared to those with SSQ score < 74.
- Example: Group 3 is 7.4 vs 11.6 and Group 4 is 8.7 vs 11.8.
- In Supplementary Table 2 if the number outside of the parenthesis is showing the median value, why are some not showing the median value?
- Example: Total SSQ Score, the median value of (58-90) is 74, the median value of (42-67) is 54.5, the median value of (81-94.8) is 88.
- Many of the values shown in Supplemental Table 2 and Table 3 are not the median values for the numbers inside the parenthesis.
- Some examples:
- The median value of Group 1 (Breads) should be 7.5 for (0-15)?
- The median value of Group 2 (Spreads) should be 3 for (0.38-6) and 5.7 for (4-7.3).
- The median value for Group 3 (Cereals,...) should be 6.2 for (4-8.3).
- The median value for Group 4 (Cheeses,...) should be 5.8 for (4-7.6).
- The median value for Group 5 (Tinned,...) should be 1.5 for (0-3).
- In Supplemental Table 2, why does Group 9 does not have a * symbol?
Author Response
Response to Reviewers Comments
- Figure 2 is duplicated in Page 6 and Page 7. The article should keep Figure 2 from Page 7, since the blue dots are important to distinguish between both groups. Are the units described in the legend for Figure 2 in Page 7 lines 209-210 correct? Green dots 24-hr urine Na≥100 mmol/d and blue dots with Na <100 mmol/L.
Response: We confirm that Figure 2 on page 7 is the updated version and the one on Page 6 is the old version which will be deleted in the clean manuscript. Thank you for spotting the unit error in the legend. It should be mmol/d and the correction has been updated.
- Figure 3 is of poor quality and cannot distinguishing the numbers. Which figures will be used in the final manuscript, top or bottom panel? Why are Fig 3B top and Fig 3B bottom different?
Response: We confirm that Figure 3 on the bottom is the correct version and the top one is the old version which will be deleted in the clean manuscript. Figure 3B has been reviewed and updated with a ROC analysis of BMI alone to predict 24-hr urine sodium excretion ≥100 mmol/day. This was completed to clarify comments to Reviewer 4.
- Figure 4 is of poor quality and cannot distinguishing the words.
Response: We have updated the Figure with enlarged texts.
- The statement of Page 9 lines 235-238 is still not clear. Other groups, besides Group 1, are also lower in persons with a SSQ≥74 compared to those with SSQ score < 74. Example: Group 3 is 7.4 vs 11.6 and Group 4 is 8.7 vs 11.8.
Response: Thank you for the comment. The sentence has been revised to include reference to other Food Categories. The revised sentence is now: “The relative percentage contribution to the total SSQ score from Group 1 (Breads) was the only category lower in the SSQ≥74 group compared to the SSQ score <74 group whereas Group 3 (cereal), Group 4 (Cheeses) and Group 6 (Processed Meat) and 7 (Flavorings) were higher (Supplementary Table 3).suggesting preferential differences in dietary sources of salt between the two groups.”
- In Supplementary Table 2 if the number outside of the parenthesis is showing the median value, why are some not showing the median value? Example: Total SSQ Score, the median value of (58-90) is 74, the median value of (42-67) is 54.5, the median value of (81-94.8) is 88.
Response: Thank you for this comment. We have checked and confirm that the numbers shown outside of parenthesis are all median values (and interquartile ranges) in Supplementary Table 2 are all correct. The distribution of the data set is skewed and therefore the median values are different to the examples provided. We have also added a legend below to the Table to describe the data for clarity.
- Many of the values shown in Supplemental Table 2 and Table 3 are not the median values for the numbers inside the parenthesis.
- Some examples:
- The median value of Group 1 (Breads) should be 7.5 for (0-15)?
- The median value of Group 2 (Spreads) should be 3 for (0.38-6) and 5.7 for (4-7.3).
- The median value for Group 3 (Cereals,...) should be 6.2 for (4-8.3).
- The median value for Group 4 (Cheeses,...) should be 5.8 for (4-7.6).
- The median value for Group 5 (Tinned,...) should be 1.5 for (0-3).
Response: We have checked and confirm that the median values and interquartile ranges in Table 2 and Table 3 are correct. The median values are different to the examples given is due to the skewness of the results in each group. We have also added a legend below to the Table to describe the data for clarity.
- In Supplemental Table 2, why does Group 9 does not have a * symbol?
Response:
Thank you for spotting this error. The * symbol has been added to Group 9 in Table 2.
Reviewer 3 Report
The paper is improved and the authors aswered to the comments.